# Construction and Evaluation of a Novel Organic Anion Transporter 1/3 CRISPR/Cas9 Double-Knockout Rat Model

**DOI:** 10.3390/pharmaceutics14112307

**Published:** 2022-10-27

**Authors:** Xueyan Gou, Fenglin Ran, Jinru Yang, Yanrong Ma, Xin’an Wu

**Affiliations:** 1Department of Pharmacy, The First Hospital of Lanzhou University, Lanzhou 730013, China; 2School of Pharmacy, Lanzhou University, Lanzhou 730000, China; 3School of First Clinical Medicine, Lanzhou University, Lanzhou 730000, China

**Keywords:** OAT1/OAT3, CRISPR/Cas9, gene knockout, pharmacokinetic

## Abstract

Background: Organic anion transporter 1 (OAT1) and OAT3 have an overlapping spectrum of substrates such that one can exert a compensatory effect when the other is dysfunctional. As a result, the knockout of either OAT1 or OAT3 is not reflected in a change in the excretion of organic anionic substrates. To date, only the mOAT1 and mOAT3 individual knockout mouse models have been available. Methods: In this study, we successfully generated a *Slc22a6/Slc22a8* double-knockout (KO) rat model using CRISPR/Cas9 technology and evaluated its biological properties. Results: The double-knockout rat model did not expression *mRNA* for rOAT1 or rOAT3 in the kidneys. Consistently, the renal excretion of *p*-aminohippuric acid (PAH), the classical substrate of OAT1/OAT3, was substantially decreased in the *Slc22a6/Slc22a8* double-knockout rats. The relative *mRNA* level of *Slco4c1* was up-regulated in KO rats. No renal pathological phenotype was evident. The renal elimination of the organic anionic drug furosemide was nearly abolished in the *Slc22a6/Slc22a8* knockout rats, but elimination of the organic cationic drug metformin was hardly affected. Conclusions: These results demonstrate that this rat model is a useful tool for investigating the functions of OAT1/OAT3 in metabolic diseases, drug metabolism and pharmacokinetics, and OATs-mediated drug interactions.

## 1. Introduction

The majority of endogenous metabolites and exogenous compounds are excreted from the body by the kidney, which is the net result of glomerular filtration, tubular secretion and reabsorption [1,2]. There are multiple transporters involved in tubular secretion, including organic anion transporters (OATs) and organic cation transporters (OCTs) [3,4], of which OAT1 and OAT3 are largely responsible for the renal uptake of organic anionic drugs and a few organic cationic drugs from the blood, and thus may be the major rate-limiting steps in the renal elimination of these drugs [5,6,7,8]. Although OAT1 and OAT3 are primarily localized in the basolateral membrane of the renal tubule [9], they are distributed in the different parts of the nephron [10]. OAT3, in addition to its overlapping distribution in the proximal tubule with OAT1, is also expressed in the thick ascending loop, the distal tubule, the connecting tubule and collecting duct [11,12]. OAT3 shares 49% sequence identity with the OAT1 gene and they have an overlapping spectrum of substrates [13,14,15,16], such as antibiotics, nonsteroidal anti-inflammatory drugs (NSAIDs), diuretics, antivirals and some endogenous compounds (uremic toxins [UTs], steroids, odorants, cyclic nucleotides, neurotransmitters, and others) [17,18]. Both OAT1 and OAT3 contribute to the basolateral uptake and renal secretion of these common substrates and work in parallel during the elimination. A lack of either OAT1 or OAT3 is compensated by the other, but not completely [19]. As a consequence, when considering the excretion of organic anionic substrates, it is necessary to consider both OAT1 and OAT3.

Recent evidence suggests that OAT1 and OAT3 not only play major roles in the elimination of exogenous compounds and endogenous substances in vivo, but also participate in the regulation of metabolism and signaling. A change in the activity of OATs will alter the elimination of metabolites mediated by these transporters [7]. Studies showed OAT1 and OAT3 are independently and synergistically involved in the disposal of 35 UTs, which were associated with inter-organ and inter-tissue communication [20]. Liu et al. reported that OAT1 participated in the metabolic pathways of the TCA cycle and the metabolism of amino acids, fatty acids, cyclic nucleotides, prostaglandins, polyamines and vitamins [7]. In addition, OAT1/3 and other SLC22 transporters have been associated with metabolic abnormalities and diseases, including chronic kidney disease, hyperuricemia, metabolic syndrome and diabetes [21,22,23,24].

Models used to study the physiological importance of OAT1 and OAT3 in vitro and vivo include cellular models (primary cells and transfected cells), renal perfusion models, kidney slice uptake models, functional inhibitor models (such as probenecid) [25,26] and gene knockout models [27]. Given the inability of the cell models to reflect the reality of the complex organisms [28,29] and the low specificity of the separated kidney models and the inhibitor models [30], the gene knockout model is a preferred option due to its high specificity and the ability to reflect the functions of transporters under physiological conditions [31]. In 2002, Nigam et al. established a mOAT3 knockout mouse model and a mOAT1 knockout mouse model in 2006 [32,33]. Although it was reported that the expression of mOAT3 also decreased in the mOAT1 knockout mouse, there was no physiologically significant effect [6,10,33]. Because of their overlapping substrate specificities, the knockout of either OAT1 or OAT3 does not truly reflect the excretion of organic anionic substrates. It is more meaningful to simultaneously knock out the OAT1 and OAT3. However, only the mOAT1 and mOAT3 knockout mouse models are available so far. Although it has been more difficult to establish knockout models in rats, they are highly suitable for pharmacodynamics, pharmacokinetics and toxicology studies [34,35,36]. As OATs are highly conserved between humans and rats, it is advantageous to study OAT1/3 functions in rats [37]. For this purpose, we developed a *Slc22a6/Slc22a8* knockout (KO) rat model to study the physiological functions of OAT1/OAT3 and their roles in disease progression.

The clustered regularly interspaced short palindromic repeats (CRISPR)–CRISPR-associated protein 9 (Cas9) system is the acquired immunity system of bacteria against phage infection and plasmid transfer [38,39]. It consists of two components forming a complex to act synergistically: single guide *RNA* (*sgRNA*) and Cas9 endonuclease [40]. Since the first successfully edited mammalian genome and human genome using CRISPR–Cas9 technology was reported in 2013 [41,42,43], the CRISPR/Cas9 system has become the primary tool for genome editing [44]. Compared with its predecessors (MN, ZFNs and TALENs), it offers high specificity, efficiency, flexibility, precision, reliability and can edit different sites simultaneously [31,41,45]. The CRISPR-Cas9 system has become the gold standard for gene editing technology and is used in various cells and organs in vivo and vitro [31,46,47].

In this study, we constructed the first rOAT1/rOAT3 double knockout rat model using CRISPR/Cas9 technology and evaluated the biological properties of this rat model. This rat model will prove to be a useful tool for investigating the functions of OAT1 and OAT3 in metabolic diseases, drug metabolism and the pharmacokinetics of drugs. In addition, it may also provide help in understanding the potential for OATs-mediated drug interactions.

## 2. Materials and Methods

### 2.1. Animals

Male and female Sprague-Dawley (SD) *Slc22a6/Slc22a8* CRISPR/Cas9 double-knockout rats were provided by Cyagen Biosciences Inc. (Guangzhou, China). The obtained animals were kept in the SPF animal room of Lanzhou University and maintained with a temperature of 24 °C, a humidity of 40% and 12-h light/dark cycles. The animals had free access to sterilized purified water and food.

The knockout targets determination, gRNA and primers design, and subsequent animal reproduction and gene identification were completed by us. The target sites were selected at the second exon of the *Slc22a8* gene and the tenth exon of the *Slc22a6* gene. The sequence of *gRNA*-A1 (matches the reverse strand of the gene) is: TAGGAATCAAAGGTCCGGATTGG and the sequence of *gRNA*-A2 (matches the forward strand of the gene) is: TCCTGTCAGTCACCCCCTGGAGG. Cyagen Biosciences Inc. was responsible for the specific implementation of knockout, and the gene identification and sequencing of several first-generation knockout animals. Firstly, the gRNA was artificially synthesized. Next, the *gRNA*-A1-Cas9 complex and *gRNA*-A2-Cas9 complex were co-injected into the fertilized rat eggs by a microinjection technique. After a few hours of incubation, the surviving fertilized eggs were transplanted into pseudo-pregnant rats to generate the target knockout offspring (Appendix A). After giving birth to the first-generation of rats, gene sequencing analysis and polymerase chain reaction (PCR) technology were used to identify the KO rats, which were bred with wild-type (WT) rats to generate the second-generation animals. In order to obtain more WT rats and homozygous rats, we raised heterozygous rats with WT rats and heterozygous rats with homozygous rats together.

### 2.2. Chemicals and Reagents

All primers for PCR/Q-PCR were synthesized by Sunbiotech Co., Ltd. (Beijing, China). The Biowest Agarose was purchased from Wobisen Technology Co., Ltd. (Beijing, China). The Nucleic Acid Stain Red was purchased from Tanon Science & Technology Co., Ltd. (Shanghai, China). The D2000 *DNA* Ladder (M1060) was purchased from Solarbio Science & Technology Co., Ltd. (Beijing, China). Phenylacetyl-L-glutamine and N-acetylcytidine were purchased from Ark Pharm, Inc. (Arlington Heights, IL, USA). Creatinine, D-kynurenine, 1-methyl-5-carboxylamide-2-pyridone and N-(cinnamoyl)glycine were purchased from Sigma-Aldrich (St Louis, MO, USA). Orotic acid was purchased from Adamas Reagent Co., Ltd. (Shanghai, China). Indole-3-acetic acid and N-(1-carboxymethyl)-L-lysine were purchased from ZZBIO Co., Ltd. (Shanghai, China). 3-Indolyl-β-D-glucopyranoside was purchased from J & K Scientific Ltd. (Beijing, China). *P*-cresol sulfate was purchased from Ziqi Biotechnology (Shanghai, China). DL-homocysteine was purchased from LVYE Biotechnology (Jiangsu, China). 3-(3, 4-Dihydroxyphenyl)-L-alanine was purchased from ACROS (Geel, Belgium). S-adenosyl-L-homocysteine, phenyl-β-D-glucuronide, 1-methyl-inosine, *p*-cresol glucuronide, 3-carboxy-4-methyl-5-propyl-2-furanpropionic acid (CMPF), 4-ethylphenyl sulfate potassium salt, N2,N2-dimethylguanosine and 3-indoxyl sulfate potassium salt were purchased from Toronto Research Chemicals (Toronto, ON, Canada). Uridine, N-acetyl-L-arginine, hippuric acid, D-neopterin, uric acid, hippuric acid, furosemide, metformin and inulin were purchased from Aladdin Reagent Co., Ltd. (Shanghai, China). Methanol (Thermo Fisher Scientific, Branchburg, NJ, USA) used throughout the study was high-performance liquid chromatography (HPLC) grade. All other reagents were of analytical grade and commercially available.

### 2.3. Genotype Identification

Total *DNA* was extracted and purified with Animal Tissue *DNA* Isolation Kit (Foregene, DE-05012) using newborn rat tails when they reached 10 days or so of age and the rats were randomly numbered by cutting the toes. After measuring the concentration of the extracted *DNA* (Nanodrop 2000, Thermo Fisher Scientific) and normalizing to the same concentration (60~120 ng/mL), the purified *DNA* was amplified (ABI, Veriti 96) using Green Taq Mix (Vazyme, P131). PCR was performed in a 25 µL volume with primers added to each reaction and the conditions as follows: 95 °C for 3 min; 95 °C for 15 s, 60 °C for 15 s, 72 °C for 60 s, 40 cycles; 72 °C for 5 min. The primers used in the genotype identification process were as follows: Primer1 (forward: 5′-CACTGTAACACTGTAATTGGTCAAAT-3′, reverse: 5′-ACAGGCTTGCCTGCAGACATTT-3′), Primer2(forward:5′-AAATGAATCTCTTCTACCTTGGTCC-3′, reverse: 5′-ACAGGCTTGCCTGCAGACATTT-3′). The amplified products were separated by electrophoresis on a 1.5% agarose gel and imaged by an automatic chemiluminescent image analysis system (Tanon 4600). The length of the targeted allele is 370 bp and the wild-type allele is 33,459 bp. If the electrophoresis results yield only one band of the size of 613 bp, it is a WT rat, and if only one band of 370 bp is evident, it is a homozygous knockout rat; if there are two bands of 613 bp and 370 bp, it is a heterozygous knock-out rat.

### 2.4. Physiological Phenotyping

We collected blood samples of 4-week-old (4W) and 7-week-old (7W) WT and KO rats through the abdominal aorta and centrifuged at 8000× *g* for 10 min. The obtained supernatant samples were sent to the Laboratory of Infectious Diseases, the First Hospital of Lanzhou University for testing. The biochemical analyses were made by a fully automated biochemical analyzer (OLYMPUS AU2700; Olympus Co., Ltd. Tokyo, Japan) according to the standards of the National Center for Clinical Laboratory (Beijing, China). Among them, the renal function indexes included urea nitrogen (BUN), creatinine (CREA), uric acid (URIC), cystatin C (CYS-C) and β2-microglobulin (β2-MG); the liver function indexes included aspartate aminotransferase (AST), alanine aminotransferase (ALT), total bilirubin (TBIL), direct bilirubin (DBIL), indirect bilirubin (IBIL), total protein (TP), albumin (ALB), globulin (GLB), alkaline phosphatase (ALP) and total bile acid (TBA); the blood lipid indexes included cholesterol (CHOL), triglyceride (TG), high-density lipoprotein cholesterol (HDL-C) and low-density lipoprotein cholesterol (LDL-C) and the blood glucose indexes—glucose (GLU). Similarly, 12-h urine samples from 4W and 7W WT and KO rats were collected and weighed and the early kidney injury markers (including microalbumin [M-ALB], CREA and N-acetyl-β-D-glucosidase [NAG]) were detected.

Kidney and liver tissues from 4W and 7W WT and KO rats were fixed in 4% paraformaldehyde, embedded in paraffin and sectioned. Sections were stained with hematoxylin-eosin (HE), Weigert iron hematoxylin-ponceau acid, fuchsin-aniline blue and toluidine blue respectively, and photographed with a light microscope (GBS, SmartWL1). To evaluate whether the renal microvessels of KO rats changed, we conducted CD31 (vascular endothelial marker) immunofluorescence staining experiments. The sectioned kidney tissues were antigen repaired with 0.1 mol/L citric acid repair solution with pH = 6 (Servicebio, G1202), blocked with serum for 1h at RT and then incubated with the CD31 antibody (1:200, Servicebio, GB113, 151) on a shaker at 4 °C overnight. The next day, the primary antibody was recovered and the fluorescent secondary antibody CY3 (1:300, Servicebio, GB21, 303) was incubated for 1h at RT in the dark. DAPI (Servicebio, G1020) was incubated in the dark for 15 min, mounted (with anti-fluorescence quenchers [Servicebio, G1401]), dried at 37 °C and photographed with a fluorescence microscope (NIKON ECLIPSE C1).

### 2.5. UTs Concentration and Renal Uptake Ratio Detection

We collected blood, 12-h urine and kidney tissue samples of 4W and 7W WT and KO rats. The concentrations of UTs were measured by the liquid chromatography-tandem mass spectrometry (LC-MS/MS), which consisted of an Agilent 1260 liquid chromatography and a 6460 triple-quadrupole mass spectrometer operated in the electrospray ionization mode (Agilent Technologies, Santa Clara, CA, USA). Chromatographic separations were performed on an Agilent ZORBAX RR StableBond C18 analytical guard column (4.6 mm × 100 mm, 3.5 µm, Agilent Technologies, Santa Clara, CA, USA) at 30 °C. The MS parameters are shown in Supporting Information Appendix A. The mobile phase of the positive ion mode compounds included methanol and water containing 0.1% formic acid (40:60, *v*/*v*) with a 0.65 mL/min flow rate and the negative ion mode compounds included acetonitrile and water (40:60, *v*/*v*) with a 0.65 mL/min flow rate. The internal standard used in the positive ion mode was d3-creatinine, and in the negative ion mode was d5-hippuric acid.

### 2.6. Compensatory Effects Detection

Total kidney *RNA* of 7W WT and KO rats was extracted with Eastep^®^ Super Total *RNA* Extraction Kit (Promega, LS1040). After determining the *RNA* concentration (Nanodrop 2000, ThermoFisher) of the samples and diluting to the same concentration (50–150 ng/ul), the extracted *RNA* was reverse transcribed into *cDNA* using RevertAid First Strand *cDNA* Synthesis Kit (Thermo Scientific, K1622) and stored at −80 °C for subsequent experiments. The reverse transcription conditions were as follows: 42 °C for 60 min and 70 °C for 5min. Real-time quantitative PCR (BIO-RAD, CFX96) was used to measure the relative *mRNA* expression of the main uptake and efflux transporters using the UltraSYBR Mixture (Low ROX) (CWBIO, CW2601). β-actin was the internal reference. The conditions were as follows: 95 °C for 10 min; 95 °C for 15 s, 60 °C for 1 min, 40 cycles. The main renal uptake transporters included *Slc22a1*, *Slc22a2*, *Slc22a7*, *Slc22a11*, *Slc22a12* and *Slco4c1*, and the efflux transporters included *Slc47a1*, *Abcc2*, *Abcc4*, *Abcb1a*, *Abcb1b* and *Abcg2*. Information on the *Slc22a6, Slc22a8* and other primers is displayed in Appendix A. The 2^−ΔΔCt^ method was used to calculate the relative *mRNA* expression.

### 2.7. Functional Evaluation and Drug Application

After the animals were fasted for 12 h and catheterized in the left femoral artery, a drug cocktail composed of PAH (5 mg/kg), metformin (5 mg/kg) and inulin (100 mg/kg) was administered by the rat tail vein to 3-month-old WT and KO rats and then blood samples were collected through the femoral artery at 5 min, 10 min, 15 min, 30 min, 45 min, 60 min, 90 min, 120 min, 180 min, 240 min, 360 min, 480 min and 600 min. The blood samples were centrifuged at 8000× *g* for 10 min and the supernatant was stored at −80 °C for detection. Similarly, after fasting for 12h and giving the same dose of drugs, urine samples of WT and KO rats were collected at 2 h, 4 h, 6 h, 8 h, 10 h, and 12 h, weighed, and stored at −80 for analysis.

After the animals were fasted for 12 h and catheterized in the left femoral artery, furosemide (20 mg/kg) [48] was orally administered to 3-month-old WT and KO rats and blood samples were collected through the femoral artery at 10 min, 20 min, 30 min, 40 min, 60 min, 90 min, 120 min, 180 min, 240 min, 360 min, 480 min, and 600 min. The blood samples were processed as described above. Similarly, after fasting for 12 h and giving the same dose of furosemide, urine samples of WT and KO rats were collected at 2 h, 4 h, 6 h, 8 h, 10 h, and 12 h, weighed, and stored at −80 for analysis.

### 2.8. Quantification Analysis

The concentrations of PAH, metformin and furosemide were determined by LC-MS/MS. The chromatographic, mass spectrometry, column and internal standard compounds were the same as described above. The MS parameters of these compounds are listed in Table 1. The mobile phase of PAH and furosemide consisted of methanol and water containing 0.1% formic acid (88:12, *v*/*v*) with a 0.65 mL/min flow rate. The mobile phase of metformin consisted of methanol and water containing 0.1% formic acid (60:40, *v*/*v*) with a 0.65 mL/min flow rate. The concentration of inulin was determined by the microplate reader method as reported by Antonin Heyrovsky in 1956 [49].

### 2.9. Statistical Data Analysis

The Drug and Statistics 2.0 program (Medical College of Wannan, Wuhu, China) was used to perform non-compartmental analyses of PAH, metformin, furosemide and inulin. All statistical analyses were performed in Prism 7 (GraphPad Software, Inc.) software using a two-tailed *t*-test. All data are presented as mean ± S.D. When the value of *p* was less than 0.05, we concluded that there is a statistically significant difference between the groups.

## 3. Results

### 3.1. Development of the Slc22a6/Slc22a8 Knockout Rat

A schematic of targeted deletion of rOAT1 and rOAT3 provided by Cyagen Biosciences Inc. is presented in Figure 1A, in which homozygous rats with 33,105-bp deletion around the second exon of the *Slc22a8* gene and the tenth exon of the *Slc22a6* gene were obtained. Rat genotypes were identified using rat tail *DNA* and specific primer sets as described in the Methods. Our results showed that the KO rats had only one band of 370 bp and the WT rats had only one band of 613 bp (Figure 1B). In addition, the relative *mRNA* levels of *Slc22a6/Slc22a8* in the kidneys of WT and KO rats were measured with *Slc22a6/Slc22a8* specific primers and Q-PCR technology. As shown in Figure 1C, compared with the WT rats, KO rats had almost no expression of *Slc22a6* and *Slc22a8* genes. Our results of the agarose electrophoresis of the amplified products also showed there were almost no amplified products in the KO rats. These results demonstrated that rOAT1/rOAT3 knockout rat model was generated.

### 3.2. Physiological Phenotyping

In order to determine whether the physiological function of the KO rats changed, we measured the serum renal function, liver function, lipids, glucose indexes and urine early renal injury makers of 4W and 7W WT and KO rats. As shown in Figure 2, there were no obvious differences in the serum biochemical indicators and urine markers at 7W between WT and KO rats. The levels of BUN, CREA (Figure 2A), NAG (Figure 2B), AST, ALT, TP, ALB, GLB, ALP and TBA (Figure 2C) in 4W KO rats were far higher than that of WT rats, the TG and Glu (Figure 2D) levels were significantly lower, and the URIC, Cys-C, β2-MG (Figure 2A), M-ALB, CREA (urine) (Figure 2B), TBIL, DBIL, IBIL (Figure 2C), CHOL, LDL-C and HDL-C (Figure 2D) levels were unaffected.

We measured the weight of the body and kidneys of 4W and 7W WT and KO rats. Compared with the WT rats, the KO rats showed no visible differences in the body weight, kidney weight, kidney/body ratio at 7W, whereas these values were significantly different at 4W (Figure 3A). To evaluate the histological structure of 4W and 7W WT and KO rats, HE staining was performed. The results indicated that the kidneys and livers of KO rats were normal in volume and no obvious histopathological changes were noted (Figure 3B,C). The Masson staining of the kidneys of 4W and 7W WT and KO rats showed both at 4W and 7W that there were no significant differences in the blue areas between the WT and KO rats (Figure 3D). The number of mast cells in the WT and KO rats was similar at 4W and 7W and there were no significant differences (Figure 3E). To evaluate whether the renal microvessels of KO rats changed, we conducted CD31 immunofluorescence staining experiments. In Figure 3F, the number of renal microvessels in 7W WT and KO rats was comparable, while in 4W WT rats the number was slightly increased over that of KO rats. These results demonstrated that the deletion of rOAT1 and rOAT3 did not cause renal inflammation or pathological changes in rat kidneys.

### 3.3. UTs Concentration and Renal Uptake Ratio Detection

In order to measure the plasma, urine concentration and renal uptake ratio of UTs, we collected blood, 12-h urine and kidney tissue samples of 4W and 7W WT and KO rats. Compared with the WT rats, the plasma concentrations of N6-(carboxymethyl)-l-lysine, N-acetyl-L-arginine, 1-methyl-inosine, uric acid, orotic acid, D-kynurenine, N-acetylcytidine, 3-indolyl-β-D-glucopyranoside, 3-indoxyl sulfate potassium salt, hippuric acid, 4-ethylphenyl sulfate and *p*-cresol sulfate of KO rats were significantly increased and the renal uptake ratios obviously decreased, among which the change of N-acetyl-L-arginine, N-acetylcytidine, 3-indoxyl sulfate potassium salt, hippuric acid, 4-ethylphenyl sulfate and *p*-cresol sulfate were the same in both 4W and 7W rats (Figure 4). On the contrary, the plasma concentrations of phenyl-β-D-glucuronide, *p*-cresol glucuronide and N-(cinnamoyl)glycine in KO rats were significantly lower than those of WT rats and the renal uptake ratios were much higher. The renal uptake ratio of N-(cinnamoyl)glycine markedly increased in the 4W and 7W rat kidneys (Appendix A). 1-Methyl-inosine, N2,N2-dimethylguanosine and 4-ethylphenyl sulfate in the 4W and 7W rat urine remained the same, of which the first two clearly decreased and 4-ethylphenyl sulfate increased (Figure 4). Comparing the 4W and 7W plasma UTs concentrations in KO rats showed that the plasma concentrations of hippuric acid, N6-(carboxymethyl)-l-lysine, 3-(3,4-dihydroxyphenyl)-L-alanine, creatinine, indole-3-acetic acid, N-acetyl-L-arginine, phenyl-β-D-glucuronide, DL-homocysteine, D-kynurenine, uridine, uric acid, orotic acid, 1-methyl-5-carboxylamide-2-pyridone, N2,N2-dimethylguanosine, 3-indoxyl sulfate potassium salt and CMPF at 4W were higher than at 7W, among which the last nine UTs had statistically significant differences (Figure 4 and Appendix A). The renal uptake ratios and urine concentrations of UTs also showed similar changes (Figure 4 and Appendix A). It can be seen that the abnormality of UTs in KO rats was more obvious at 4W than that at 7W.

### 3.4. Detection of Compensatory Effects

To determine if there were compensatory changes in the expression of other major renal transporters in KO rats, we used Q-PCR to detect relative *mRNA* levels. As shown in Figure 5, compared with the WT rats, the *mRNA* expression of *Slc22a11* and *Abcc4* were down-regulated in the kidneys of KO rats. The *mRNA* expression of *Slco4c1* was up-regulated in the kidneys of KO rats, while the expression of *Slc22a7*, *Slc22a1*, *Slc22a2*, *Slc47a1*, *Abcc2*, *Abcb1a*, *Abcb1b*, *Slc22a12* and *Abcg2* was unchanged. This suggested that when looking for polymorphisms associated with OAT1 and OAT3 responsiveness, we might need to simultaneously consider the *Slc22a6* and *Slc22a8* as well as other genes. Furthermore, we used the amplification products of Q-PCR to perform agarose gel electrophoresis to verify the specificities of the primers. The results demonstrated that all the selected primers only had one band, which confirmed that the primers had high specificities (Appendix A).

### 3.5. Functional Evaluation and Drug Application

To verify that rOAT1/3 functions were missing in KO rats, we studied the change in the pharmacokinetic and urinary excretion behavior of PAH. Furthermore, we measured the pharmacokinetic and urinary excretion characteristics of inulin to evaluate the glomerular filtration rate (GFR) of KO rats. Compared with the WT rats, the plasma concentrations of PAH and inulin were much higher in KO rats (Figure 6A,B) and the urinary excretions were significantly decreased (Figure 6C,D). Consistently, the area under the time–plasma concentration curve (AUC) and maximum concentration (Cmax) of PAH and inulin were substantially increased in KO rats and the apparent volume of distribution/bioavailability (V1/F), and clearance/bioavailability (CL/F) significantly decreased (Table 2).

After evaluating the pharmacokinetic and urinary excretion behavior of PAH, we investigated the pharmacokinetic and urinary excretion profiles of the anionic drug, furosemide and the cationic drug, metformin. Compared with the WT rats, the plasma furosemide concentration of KO rats was significantly increased at 20, 30, 40, 60, 90, 120, 180, 240, 360, 480 and 600 min (Figure 7A) and the urine furosemide excretion was markedly decreased at 2, 4, 6, 8, 10, 12 h (Figure 7B). At the same time, the AUC(0–t), AUC(0–∞), Cmax, time to reach maximum concentration (Tmax) and absorption rate constant (Ka) of KO rats were obviously increased by 526.9%, 444.3%, 183.2%, 190% and 536.7% respectively, while the V1/F and CL/F were clearly reduced by 378.5% and 591.8% (Table 3). The mean residence time (MRT) showed no visible difference. The plasma concentration of metformin in KO rats was significantly increased (Figure 7C), but there were no obvious differences in the urine concentration (Figure 7D). The pharmacokinetic parameters of metformin such as the AUC(0–t), AUC(0–∞), Cmax and CL/F were significantly different between the WT and KO rats (Table 3). Therefore, the renal elimination of the anionic drug furosemide was almost abolished after knocking out rOAT1/rOAT3, and the excretion of the cationic drug metformin was hardly affected.

## 4. Discussion

There has been widespread attention paid to the physiological importance of OAT1 and OAT3. They play vital roles in metabolism and signaling in addition to their traditional renal uptake functions. It is essential to establish a precise knockout animal model to study their physiological functions of these two transporters. Although Eraly and Sweet constructed mOAT1 and mOAT3 knockout mouse models by homologous recombination technology [32,33], rats offer significant advantages over mouse and CRISPR technology can edit different sites simultaneously [34,35,36]. As shown in the NCBI database, *Slc22a6*, *Slc22a8*, and LOC102550957 genes of rats are all located in the 1q43 of the chromosome, and the gene positions of LOC102550957 and *Slc22a6/8* partially overlap. Whether knocking out OAT1 and OAT3 separately or both at the same time, it is unavoidable to delete part of LOC102550957 gene. Previous investigators knocked out either *Slc22a6* or *Slc22a8* and used probenecid to inhibit the remaining transporter so as to mimic a double-knockout of *Slc22a6*/*8* [20]. However, probenecid is a non-specific inhibitor [50,51], and may inhibit other transporters while inhibiting the OATs. In view of the difficulty of deleting both genes simultaneously, we constructed the rOAT1/rOAT3 double-knockout rat model by deletion of the majority of exons and a LOC102550957 gene between *Slc22a6* and *Slc22a8*. LOC102550957 belongs to *lncRNA*, which do not encode proteins. In addition, there is a LOC102550957 gene between the rat *Slc22a6* and *Slc22a8* genes in the NCBI database, while no other genes in the Ensembl database. Since this gene is not shared by the two databases and its function has not been reported in the literature, the LOC102550957 gene may not have a function. Importantly, our results of pharmacokinetics and urinary excretion behaviors of furosemide and PAH were identical to those obtained by Eraly et al. [33], indicating that the phenotype of KO rats can be wholly attributed to the knockout of *Slc22a6* and *Slc22a8*, and the LOC102550957 gene between appears not have a function. Therefore, our approach of constructing the rOAT1/rOAT3 double knockout rat model is clearly preferable to these studies. The knockout animals (both male and female) grew and developed normally, appeared healthy, and had normal lifespans. This was consistent with the performance of the mOAT1 knockout mouse generated by Liu et al. [7]. In conclusion, although some gene fragments were deleted during double gene knockout, there is no reproductive toxicity and obvious liver and kidney injury, making that this model is very suitable to research the renal excretion of organic anionic drugs.

In order to verify that the *Slc22a6/Slc22a8* double-knockout model was successfully constructed, we investigated the *DNA* structure, *mRNA* expression and functional performance of KO rats. As shown in Figure 1C, there were a few PCR products in the KO animals. The results may be due to the weak primer specificity. Our previous low specificity primer results displayed that there was no amplification product of *Slc22a8* in agarose gel electrophoresis of KO rats (Appendix A). It showed that although the specificity of the primer was low, it was sufficient to prove that there was no amplification product of *Slc22a8* in KO rats, which indicate that the deletion of *Slc22a8* was successful. It is worth noting that although the cumulative urinary excretion of PAH was significantly decreased in KO rats, there was still some urinary excretion. Momper et al. reported the PAH had the characteristics of being mainly transported and secreted by OAT1/3 in the renal tubule, free glomerular filtration, low plasma protein binding rate and no tubular reabsorption [17,52]. Therefore, the urinary excretion of PAH in our KO rats was mainly due to the glomerular filtration of PAH. In short, a *Slc22a6/Slc22a8* double knockout model was successfully constructed.

A normal physiological state for experimental animals is a prerequisite for eliminating interference and ensuring reliable experimental results [36]. Results of biochemical indexes and UTs both indicated that the abnormality of physiological function in KO rats was more obvious at 4W than that at 7W. Transcripts of OAT1 and OAT3 appear in the renal proximal tubule during late pregnancy, and their expression gradually increases after birth, peaks in adulthood and remains at a high level thereafter [53,54]. The different physiological states of KO rats at 4W and 7W might be caused by their diverse developmental expressions. We thought that the differences in renal blood vessels and the body’s self-regulation and compensatory mechanisms might be also reasons for the differences in biochemical indexes between 4W and 7W rats, but this requires further research. Since the physiological state of KO rats after 7W was relatively stable, it is best to use rats after 7W for research. Importantly, the kidney function of KO rats at 12 weeks was similar to that at 7 weeks, and these rats were more suitable to carry out pharmacokinetic research for their large size and blood volume. Since inulin had the following characteristics: (1) not metabolized in vivo; (2) excreted only by glomerular filtration; (3) not bound to proteins and freely filtered by the glomerular capillary membrane; (4) easy to quantify with high accuracy and generally considered to be the standard reference substance for GFR [55], we used inulin to evaluate GFR of KO rats. The pharmacokinetics and urinary excretion behavior of inulin indicated that the knockout of *Slc22a6/Slc22a8* reduced the GFR of KO rats. Taken together, although the deletion of these vital transporters, *Slc22a6/Slc22a8* double-knockout rats were free of morphological and lifespan abnormalities.

UTs are a group of molecules eliminated by the kidneys that often accumulate in the plasma of patients with renal insufficiency [56]. Previous studies showed that many UTs interacted with OAT1 and OAT3 in vivo, thereby regulating UTs metabolism independently and cooperatively and governing the excretion of UTs and their metabolites from the body [21,37,57,58]. The plasma levels of N6-(carboxymethyl)-l-lysine, N-acetyl-L-arginine, 1-methyl-inosine, uric acid, orotic acid, D-kynurenine, N-acetylcytidine, 3-indolyl-β-D-glucopyranoside, 3-indoxyl sulfate potassium salt, hippuric acid, 4-ethylphenyl sulfate and *p*-cresol sulfate in KO rats significantly increased and the renal uptake ratios markedly decreased compared with WT rats. These results are consistent with the findings that creatinine, uric acid, 3-indoxyl, sulfate potassium salt, hippuric acid and *p*-cresol sulfate are common substrates of OAT1 and OAT3, while the kynurenine, orotic acid and N2, N2-dimethylguanosine are substrates of OAT1 and CMPF is a substrate of OAT3 [20]. Al-though it has been shown that creatinine is a common substrate of OAT1 and OAT3, its plasma level was not significantly increased in the double KO rats, which may be because creatinine is also transported into the kidney by OCT2 in addition to OAT1 and OAT3. This can be seen from the study by Topletz-Erickson et al. [59]. In addition, our findings indicate that N-acetyl-L-arginine, N6-(carboxymethyl)-l-lysine, 3-indolyl-β-D-glucopyranoside, 1-methyl-inosine, N-acetylcytidine, 4-ethylphenyl sulfate might also be substrates of OAT1 and/or OAT3 but require further validation using uptake experiments in OAT1 and OAT3 overexpressing cells. Overall, the plasma and urine UTs concentrations and renal uptake ratios of UTs in KO rats changed at 4W or 7W, which is expected to reflect the processing changes due to the rOAT1 and rOAT3 deficiency.

After validating the KO model, we investigated the effects of the knockout of *Slc22a6/Slc22a8* genes on the excretion behavior of anionic drug furosemide and the cationic drug metformin [60,61]. The plasma concentration of furosemide was significantly higher in KO rats than in WT rats, and the cumulative urinary excretions was nearly abrogated. In contrast to the urinary excretion behavior of PAH, furosemide showed almost no urinary excretion. This might be due to the very high plasma protein binding rate of furosemide, which is not filtered by the glomerulus and can only be excreted by the proximal tubular secretion [19]. The plasma concentration of metformin in KO rats was slightly increased compared with the WT rats, but there was no obvious difference in the urinary concentration. The *mRNA* levels of rOCTs that mediate metformin uptake into the kidney were not affected by double knockout. Therefore, this change in metformin plasma levels in KO rats was not caused by OCTs. The knockout of the *Slc22a6/Slc22a8* gene resulted in the plasma accumulation of numerous UTs and decreased urinary elimination, which caused the blood osmotic pressure to be higher than that of the urine. The elevated plasma concentration of metformin could be attributed to the higher plasma osmotic pressure of KO rats or tissue distribution differences of metformin in KO rats.

## 5. Conclusions

We used CRISPR/Cas9 technology to construct a *Slc22a6/Slc22a8* double-knockout rat model with no renal pathological phenotype firstly. The renal elimination of the anionic drug furosemide was almost abolished by the rOAT1/rOAT3 deficiency, and there was no compensation by other types of tubular secretion. This rat model is a useful tool for investigating the functions of rOAT1 and rOAT3 in metabolic diseases, drug metabolism and pharmacokinetics, as well as OATs-mediated drug interactions.

## Figures and Tables

**Figure 1 pharmaceutics-14-02307-f001:**
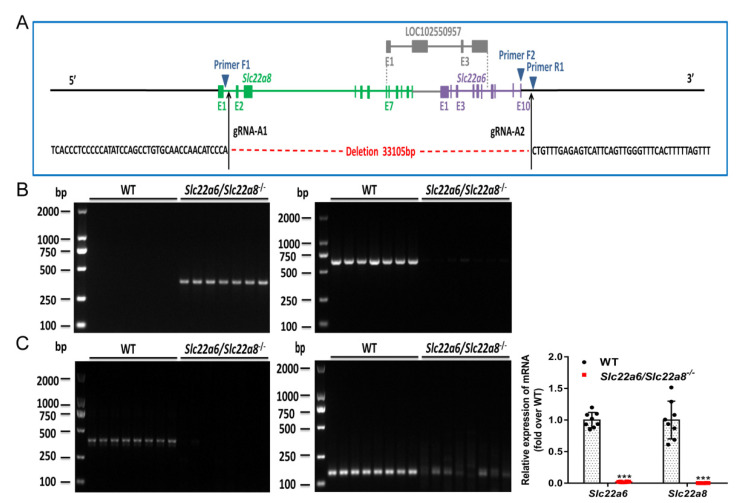
Development and validation of the *Slc22a6/Slc22a8* double-knockout model. PCR and Q-PCR were used to verify the *Slc22a6/Slc22a8* knockout model. (**A**) Schematic of the targeted deletion of *Slc22a6* and *Slc22a8*. The black arrows indicate the target sites. The positions of the PCR primers used to detect the wild-type allele and targeted allele are shown with blue triangles. (**B**) Agarose gel electrophoresis of the PCR Primers1 amplification product with 370 bp targeted allele (**left**) and PCR Primers2 amplification product with 613 bp wild-type allele (**right**) (n = 7). (**C**) Agarose gel electrophoresis of the Q-PCR amplification products of *Slc22a6* (**left**) and *Slc22a8* (**middle**) primers (n = 8). The *mRNA* expression of *Slc22a6* and *Slc22a8* in the kidneys of WT and KO rats (right). *** *p* < 0.001 compared with WT rats.

**Figure 2 pharmaceutics-14-02307-f002:**
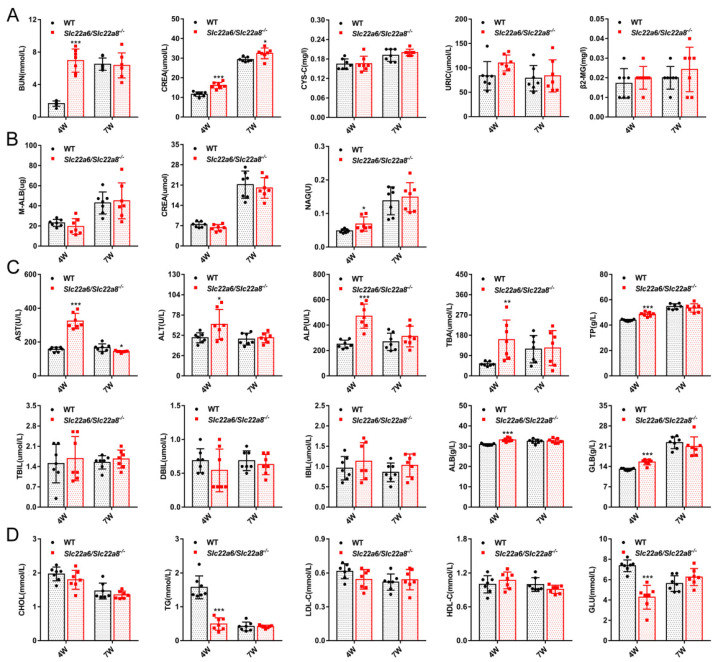
Effects of *Slc22a6/Slc22a8* double-knockout on the biochemical indicators in 4W and 7W WT and KO rats. Biochemical indicators include renal function, liver function, blood lipids and glucose indexes. (**A**) Serum renal function indicators BUN, CREA, CYS-C, URIC and β2-MG. (**B**) Urine early kidney injury markers M-ALB, CREA and NAG. (**C**) Serum liver function indicators AST, ALT, ALP, TBA, TP, TBIL, DBIL, IBIL, ALB and GLB. (**D**) Serum lipids CHOL, TG, LDL-C, HDL-C and glucose indicators. Data are presented as mean ± SD (n = 7). *, *p* < 0.05; **, *p* < 0.01; ***, *p* < 0.001 compared with WT rats.

**Figure 3 pharmaceutics-14-02307-f003:**
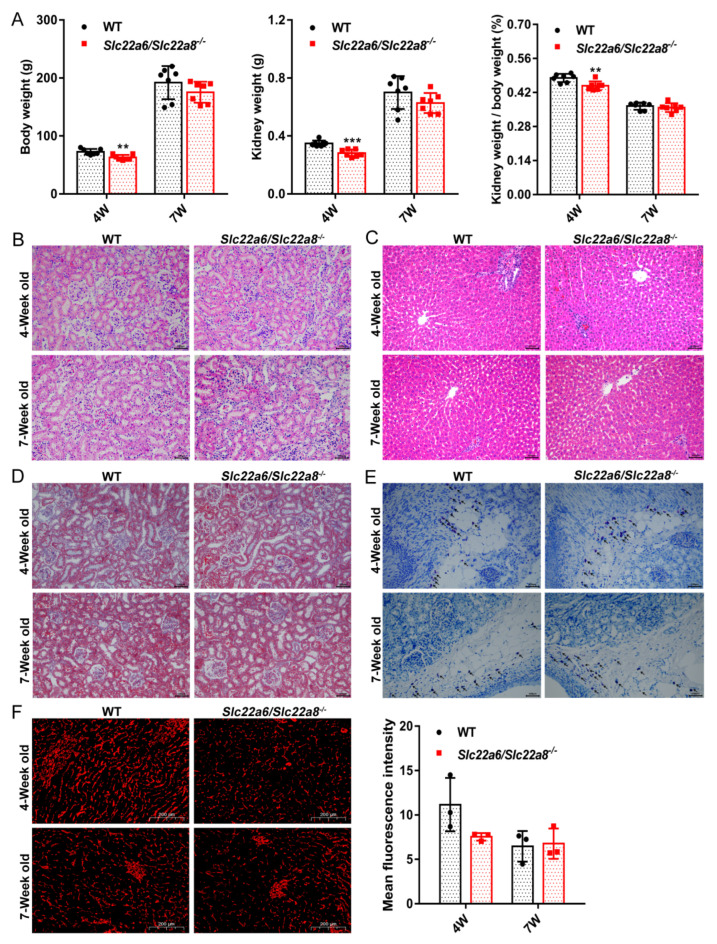
Effects of *Slc22a6/Slc22a8* double-knockout on the body and kidney weights, physiological phenotype of the kidneys and livers in 4W and 7W WT and KO rats. (**A**) Body weight, kidney weight and kidney weight/body weight of 4W and 7W WT and KO rats (n = 7). (**B**) Hematoxylin and Eosin staining section of the kidneys in 4W and 7W WT and KO rats (200×) (n = 3). The cytoplasm was stained mauve by eosin and the cell nucleus was stained blue by hematoxylin. (**C**) Hematoxylin and Eosin staining section of the livers in 4W and 7W WT and KO rats (200×) (n = 3). (**D**) Masson staining section of the kidneys in 4W and 7W WT and KO rats (200×) (n = 3). The collagen fibers were stained blue. (**E**) Toluidine blue staining section of the kidneys in 4W and 7W WT and KO rats (200×) (n = 3). Mast cells were stained purple. Positive results are indicated by arrows. (**F**) Immunohistochemistry analysis of CD31 protein expression in the kidneys of 4W and 7W WT and KO rats (5×) (n = 3). The scale bar in picture (**B**–**E**) was 100 μm in length and the picture (**F**) was 200 μm. **, *p* < 0.01; ***, *p* < 0.001 compared with WT rats.

**Figure 4 pharmaceutics-14-02307-f004:**
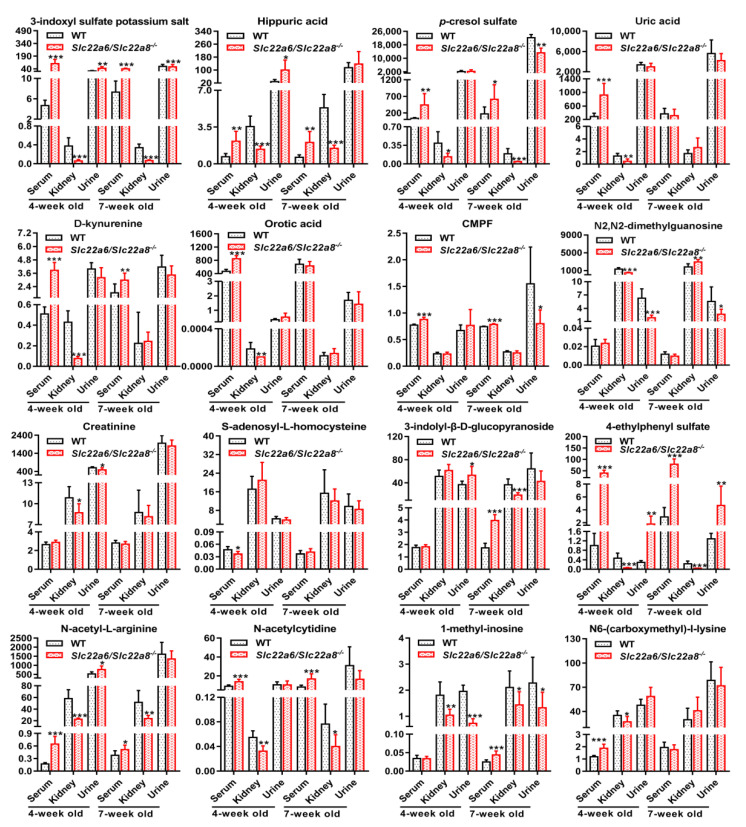
Effects of *Slc22a6*/*Slc22a8* double-knockout on the endogenous UTs (3-indoxyl sulfate potassium salt, hippuric acid, *p*-cresol sulfate, uric acid, D-kynurenine, orotic acid, CMPF, N2,N2-dimethylguanosine, creatinine, S-adenosyl-L-homocysteine, 4-ethylphenyl sulfate, 3-indolyl-β-D-glucopyranoside, N-acetyl-L-arginine, N-acetylcytidine, 1-methyl-inosine, N6-(carboxymethyl)-l-lysine) in 4W and 7W WT and KO rats. Plasma, urine concentration and kidney uptake ratio of UTs in 4W and 7W WT and KO rats. The unit of the plasma and urine concentration of UTs is ug/mL and the renal uptake ratio is ml/g. Data are presented as mean ± SD (n = 7). *, *p* < 0.05; **, *p* < 0.01; ***, *p* < 0.001 compared with WT rats.

**Figure 5 pharmaceutics-14-02307-f005:**
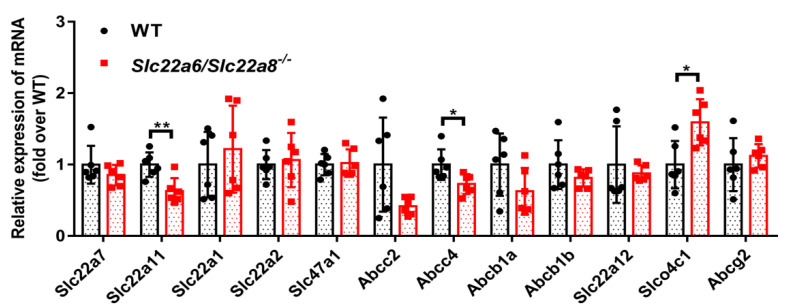
Effects of *Slc22a6*/*Slc22a8* double-knockout on the *mRNA* expression of major transporters in the kidneys of WT and KO rats. Q-PCR was used to detect the *mRNA* expression of the transporters. Among them, *Slc22a7*, *Slc22a11*, *Slc22a1*, *Slc22a2*, *Slc22a12* and *Slco4c1* are uptake transporters, and *Slc47a1*, *Abcc2*, *Abcc4*, *Abcb1a*, *Abcb1b*, *Abcg2* are efflux transporters. Data are presented as mean ± SD (n = 6). *, *p* < 0.05; **, *p* < 0.01 compared with WT rats.

**Figure 6 pharmaceutics-14-02307-f006:**
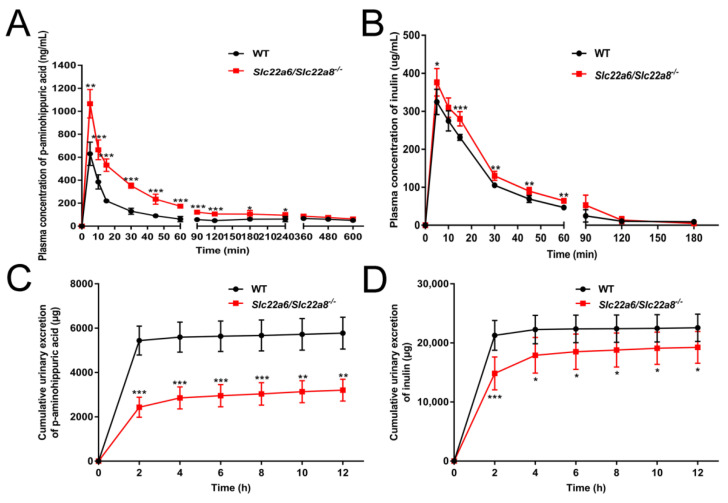
Effects of *Slc22a6*/*Slc22a8* double-knockout on the pharmacokinetics and urinary excretions of substrate of OAT1/OAT3 and inulin. (**A**) Plasma concentration-time curve of PAH (n = 5). (**B**) Plasma concentration-time curve of inulin (n = 5). (**C**) Cumulative urinary excretion-time curve of PAH (n = 6). (**D**) Cumulative urinary excretion–time curve of inulin (n = 6). Data are presented as mean ± SD. *, *p* < 0.05; **, *p* < 0.01; ***, *p* < 0.001 compared with WT rats.

**Figure 7 pharmaceutics-14-02307-f007:**
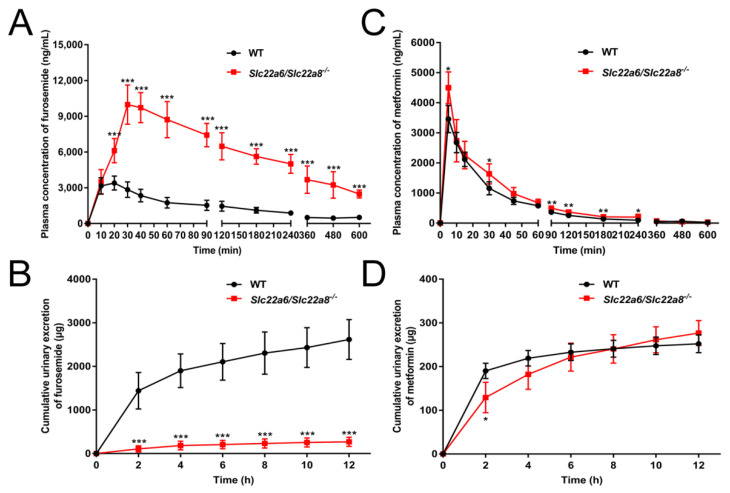
Effects of *Slc22a6/Slc22a8* double-knockout on the pharmacokinetics and urinary excretions of the anionic drug furosemide and the cationic drug metformin. (**A**) Plasma concentration-time curve of furosemide (n = 6). (**B**) Cumulative urinary excretion-time curve of furosemide (n = 7). (**C**) Plasma concentration-time curve of metformin (n = 5). (**D**) Cumulative urinary excretion-time curve of metformin (n = 6). Data are presented as mean ± SD. *, *p* < 0.05; **, *p* < 0.01; ***, *p* < 0.001 compared with WT rats.

**Table 1 pharmaceutics-14-02307-t001:** Mass spectrometric parameters of PAH, metformin, furosemide and internal standard.

Name	Precursor Ion (*m*/*z*)	Product Ion (*m*/*z*)	Fragmentor /V	Collision Energy/V	Polarity
PAH	193	92	80	12	negative
metformin	130.1	60.2	80	10	positive
furosemide	329	285	110	12	negative
d3-creatinine	117	47.2	70	16	positive
d5-hippuric acid	183.4	139.3	80	6	negative

PAH: *p*-aminohippuric acid.

**Table 2 pharmaceutics-14-02307-t002:** Pharmacokinetic parameters of PAH and inulin after intravenous injection in rats.

PK Parameters	WT	*Slc22a6/Slc22a8^-/-^*
A. PAH		
AUC(0–12) (ng/mL·min)	40,246.44 ± 6051.31	76,878.23 ± 6318.77 ***
AUC(0–∞) (ng/mL·min)	100,002.31 ± 32,256.69	140,971.46 ± 38,041.04
MRT(0–12) (min)	212.48 ± 35.37	197.30 ± 21.00
MRT(0–∞) (min)	766.23 ± 191.25	602.93 ± 94.80
t1/2α (min)	9.03 ± 1.72	14.78 ± 2.39 **
Cmax(ng/mL)	630.13 ± 102.16	993.31 ± 194.47 **
V1/F (L/kg)	6,767,614.82 ± 1,280,508.60	4,829,527.24 ± 615207.36 *
CL/F (L/min/kg)	53,876.26 ± 15,666.85	37,165.61 ± 7984.62
B. Inulin		
AUC(0–12) (ng/mL·min)	11,386.00 ± 1048.99	14,208.11 ± 1433.67 **
AUC(0–∞) (ng/mL·min)	13,563.90 ± 1282.57	16,434.89 ± 1696.15 *
MRT(0–12) (min)	28.71 ± 5.54	35.34 ± 4.08
MRT(0–∞) (min)	34.03 ± 8.70	37.99 ± 4.20
t1/2α (min)	14.75 ± 4.72	5.23 ± 2.51 **
Cmax (ng/mL)	324.47 ± 33.41	376.40 ± 36.13 *
V1/F (L/kg)	241,341.96 ± 36672.44	185,957.58 ± 15,232.62 *
CL/F (L/min/kg)	7427.26 ± 724.03	6137.27 ± 639.31 *

AUC(0–12), area under the plasma concentration-time curve from time zero to 12 h; AUC(0–∞), area under the plasma concentration-time curve from time zero extrapolated to infinite time; MRT, mean retention time; t1/2α, absorption half life; Cmax, maximum plasma concentration; V, apparent volume of distribution; F, bioavailability; CL, clearance. Data are expressed as mean ± S.D. *, *p* < 0.05; **, *p* < 0.01; ***, *p* < 0.001 compared with WT rats, n = 5.

**Table 3 pharmaceutics-14-02307-t003:** Pharmacokinetic parameters of furosemide and metformin after administration in rats.

PK Parameters	WT	*Slc22a6/Slc22a8^-/-^*
A. Furosemide		
AUC(0–12) (ng/mL·min)	415,149.60 ± 107,269.39	2,602,712.48 ± 398,597.50 ***
AUC(0–∞) (ng/mL·min)	836,411.53 ± 427,145.03	4,552,421.65 ± 2,973,211.06 *
MRT(0–12) (min)	200.67 ± 24.87	205.26 ± 75.68
MRT(0–∞) (min)	558.24 ± 304.79	474.44 ± 238.44
t1/2α (min)	69.32 ± 0	61.37 ± 19.47
t1/2Ka (min)	69.32 ± 0	17.40 ± 11.22 ***
Tmax (min)	16.67 ± 5.16	48.33 ± 23.17 **
Cmax (ng/mL)	3613.94 ± 577.61	10,235.97 ± 1310.57 ***
V1/F (L/kg)	11,783,245.28 ± 4,611,552.51	2,462,792.44 ± 837,625.09 ***
CL/F (L/min/kg)	30,173.78 ± 15,894.35	4361.74 ± 1521.40 **
Ka (1/min)	0.01 ± 0	0.05 ± 0.03 **
B. Metformin		
AUC(0–12) (ng/mL·min)	159,175.91 ± 16,583.51	208,059.11 ± 25,719.35 **
AUC(0–∞) (ng/mL·min)	181,705.33 ± 17,121.65	239,759.06 ± 28,290.82 **
MRT(0–12) (min)	81.69 ± 8.84	91.92 ± 14.95
MRT(0–∞) (min)	88.71 ± 3.63	114.92 ± 29.87
Cmax (ng/mL)	3458.21 ± 451.59	4498.47 ± 527.50 *
V1/F (L/kg)	1,355,217.65 ± 136,727.23	1,278,640.28 ± 336,921.11
CL/F (L/min/kg)	27,715.97 ± 2643.84	21,090.99 ± 2516.98 **

Tmax, time to reach maximum concentration; Ka, absorption rate constant; Data are expressed as mean ± S.D. *, *p* < 0.05; **, *p* < 0.01; ***, *p* < 0.001 compared with WT rats, furosemide, n = 6; metformin, n = 5.

## Data Availability

The datasets generated for this study are available from the corresponding author on reasonable request.

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
