# Peer review of "Construction and Evaluation of a Novel Organic Anion Transporter 1/3 CRISPR/Cas9 Double-Knockout Rat Model"

_pharmaceutics, 2022, doi:10.3390/pharmaceutics14112307_

Round 1

Reviewer 1 Report

The authors present the case that the Oat1/Oat2 (Slc22a6/Slc22a8) KO model will be a good tool for understanding metabolism and anion transport and excretion.  I believe that they will be right, and that it will be a better model than the same mutation(s) in the mouse.  However:  the authors did not compare the double KO with each single KO, and the mutation was created by deletion of the majority of exons and the intergenic region between Slc22a6 and Slc22a8.  It is not a given that there isn't some other regulatory element in the intergenic region that may have been deleted that contributes to the phenotype.  It is important to show, not by analogy to the mouse, but by direct comparison, the individual effects of single Oat1 KO and Oat2 KO in the rat and the double KO that eliminates the presumed redundancy.

Reviewer 2 Report

Review of the manuscript titled „Construction and Evaluation of an Organic Anion Transporter 1/3 CRISPR/Cas9 Double Knock-out Rat Model” by Gou and colleagues submitted to Pharmaceutics.

The manuscript describes the generation and detailed characterization of the first double, Slc22a6/Slc22a8 knock-out (KO) rat model. Since both proteins determine renal clearance of (mostly) anionic drugs, they are recommended for investigation during drug development by international regulatory agencies (e.g., FDA). Therefore, the model developed by the authors can be a useful tool to investigate the involvement of these transporters in drug metabolism in vivo. Moreover, if I understand correctly, this is the very first double KO model, however, the authors failed to emphasize this.

Still, I have few major comments, and also, the manuscript needs significant language editing.

Major comments:

1.      Lane 29: “The majority of endogenous metabolites and exogenous compounds rely on the kidney to excrete from the body, which is a net result of glomerular filtration, tubular secretion and reabsorption” The authors forgot to mention the liver, that has even more importance in detoxification.

2.      Lane 64: The “inhibitor model” for studying OATs is not defined

3.      In contrast to its title, Table 1 does not contain data about 5-fluorouracil

4.      Sequences of the primers used to detect Slc22a6 or Slc22a8 (Figure 1C) is not disclosed anywhere in the manuscript. The authors do not explain the presence of PCR products in Figure 1C right panel in the KO animals.

5.      Figure 4: Since all uremic toxins are shown on the same graph, some data are hardly visible and hence, the reviewer cannot evaluate the difference between the WT and KO rats for each compound (even though the asterisk indicate some statistical significance). Redrawing of the graph is suggested.

6.      The authors showed that 4 week-old rats have increased renal impairment compared to 7 week-old or wild-type rats (e.g., Figure 2), that they explain by adaptation of the animals to renal disfunction, e.g. by changed expression of other transporters. Still, PAH and furosemide pharmacokientics was examined in 3 moth old rats. The authors should explain this. There is some mention about this in Discussion, Lane 512-513: “it is best to use the rats after 512 7W for research.” I find this not scientific enough.

7.      Discussion is repetitive and overlapping with the Results section.

Round 2

Reviewer 1 Report

After consideration of results published by Wu et al. (ref. #20), I agree with the authors' response that "considering previous studies and what is now known about these two very similar transporters, the construction of two single OAT-KO models would entail significant work while providing little new knowledge; the double knockout most clearly advances our knowledge."  While CRISPR offers the ability to create the individual KO alleles, it is not clear that the time and effort required would provide information that would affect the authors' conclusions.

The authors have addressed the concern about deletion of potential elements at the locus, particularly LOC102550957, in the discussion.  lncRNAs may not encode proteins, but certainly may encode regulatory RNAs.  While it is possible that this element has no function, the authors may suggest but cannot conclude that it does not based on the lack of observable phenotype.  Therefore, I recommend that they adjust the discussion accordingly.

One minor point:  because CRISPR is now considered standard technology, the paragraph describing CRISPR at the end of the introduction can be omitted.  The technique is adequately described in the Methods section.